# Surrogate-Assisted Differential Evolution for the Design of Multimode Resonator Topology

**DOI:** 10.3390/s24155057

**Published:** 2024-08-05

**Authors:** Vladimir Stanovov, Sergey Khodenkov, Sergey Gorbunov, Ivan Rozhnov, Lev Kazakovtsev

**Affiliations:** 1Institute of Informatics and Telecommunications, Reshetnev Siberian State University of Science and Technology, Krasnoyarsk 660037, Russia; hsa1982sibsau@mail.ru (S.K.); ris2005@mail.ru (I.R.); levk@bk.ru (L.K.); 2School of Space and Information Technology, Siberian Federal University, Krasnoyarsk 660041, Russia; jee1ary@yandex.ru

**Keywords:** multimode resonator, amplitude–frequency characteristics, microwave sensor, optimization, differential evolution, surrogate-assisted optimization, kriging, Gaussian process

## Abstract

The microstrip devices based on multimode resonators represent a class of electromagnetic microwave devices, promising use in tropospheric communication, radar, and navigation systems. The design of wideband bandpass filters, diplexers, and multiplexers with required frequency-selective properties, i.e., bandpass filters, is a complex problem, as electrodynamic modeling is a time-consuming and computationally intensive process. Various planar microstrip resonator topologies can be developed, differing in their topology type, and the search for high-quality structures with unique frequency-selective properties is an important research direction. In this study, we propose an approach for performing an automated search for multimode resonators’ conductor topology parameters using a combination of evolutionary computation approach and surrogate modeling. In particular, a variant of differential evolution optimizer is applied, and the model of the target function landscape is built using Gaussian processes. At every iteration of the algorithm, the model is used to search for new high-quality solutions. In addition, a general approach for target function formulation is presented and applied in the proposed approach. The experiments with two microwave filters have demonstrated that the proposed algorithm is capable of solving the problem of tuning two types of topologies, namely three-mode resonators and six-mode resonators, to the required parameters, and the application of surrogated-assisted algorithm has significantly improved overall performance.

## 1. Introduction

Microwave devices are widely used in many areas of communication and sensor networks. In particular, high-frequency selective devices based on microstrip resonators are characterized by a low cost in mass production and high-quality features [1]. They can also be applied as sensors for measuring the electrical properties of materials [2,3]. However, their main application is in the field of telecommunication, both tropospheric and space, as well as navigation [4]. Different types of frequency-selective devices include bandpass filters, diplexers, multiplexers, and power dividers [5,6].

The traditional application field, which uses radio equipment, requires new promising designs, as the development of high-quality electronics relies on the continuous improvement of electrical characteristics and hence opens possibilities in size reduction [4,7]. Therefore, studying the space of possible topologies of microstrip resonators is an important research field. The design features of the strip conductor allow for bringing together several lower oscillation modes in frequencies, which form a bandwidth [8]. Based on the resonators it is possible to implement filters, including both bandpass and notch (stop band) filters [9,10]. However, the microwave sensor design is also of particular interest [11,12]. The idea here is to use the changes in frequency response based on dielectric permittivity [13,14,15,16]. These designs can be used to perform dielectric measurements of solids and liquids [17,18], as well as gas sensing [12] and noninvasive medical measurements [19,20].

The two-mode [1] and three-mode designs [21] are relatively simple, which, however, allow to perform frequency-selective tasks with high efficiency if properly tuned. Due to the small number of links in such devices, their size is relatively small, as well as the passband losses. The strip conductors usually have a rectangular form or a combination of rectangles. In a bandpass filter, the amount of coupling required between the resonances is created by etching one of the conductor corners in the shape of a square, or by adding a segment of a slotted line, or a strip coupling element, which are usually located at a 45 degree angle to the resonator axes [1].

The possible requirements for resonators could be different, for example, in some cases, it is important to cover a wide range of frequencies, and in other cases, it is important to obtain low losses. Due to their small size, microstrip resonators could be applied in small wireless networking devices, including radar sensors [22,23] and antenna sensors [2]. However, it requires very accurate tuning and manufacturing of the devices—small changes in the size of the construction may significantly change the properties. Hence, here, we consider the problem of tuning the resonators to the required properties using software electrodynamic analysis tools. The capabilities of modern computing systems enable us to perform electrodynamic analysis many times while changing the geometric parameters of the microwave device.

The tuning and design of resonator topologies is a computationally expensive optimization problem due to the need to run simulations for every solution. There are various optimization techniques developed for expensive problems, for example, the surrogate approach, in which a model of the target function landscape is built [24]. Based on the usage of Gaussian processes, which are also called kriging [25], efficient global optimization (EGO) was proposed [26]. The EGO algorithm has found many applications in real-world, expensive optimization tools.

As the properties of the optimization problem to be solved are not known beforehand, and the derivative information is not available, searching for an optimal topology can be considered a black-box optimization task. A large variety of efficient black-box optimizers have been proposed in the area of evolutionary computation (EC) [27,28], including genetic algorithms (GAs) [29], particle swarm optimization (PSO) [30], and evolutionary strategies (ESs) [31]. However, in the recent decade, differential evolution (DE) has become one of the most widely used numerical optimizers due to its simplicity and high efficiency [32]. EC methods have also found many real-world applications [33].

The focus of this study is on combining the advantages of both evolutionary algorithms and surrogate-based optimization methods in order to solve a complex problem of multimode resonator design. In particular, a recently proposed variant of the differential evolution, L-SRDE, is combined with kriging in a local area around the best-known solution. The experiments are performed with two types of constructions, and the comparison results show that the proposed hybrid surrogate-assisted approach significantly outperforms both the differential evolution and the surrogated-based optimization.

The rest of this paper is organized as follows. The second section describes the microstrip resonators, their topologies, and their properties, as well as the DE and surrogate-based optimization algorithms. The third section contains a description of the proposed approach to automated microwave device tuning. The experimental set-up and results are shown in the third section, and the last section concludes this study.

## 2. Materials and Methods

### 2.1. Microstrip Resonators

Microstrip resonators are components of selection devices that are used in different types of communication devices and are mainly characterized by their AFC. In this study, planar resonators are considered, in particular, two constructions. The first one is U-shaped (slot-split rectangular), and the second one has U-shape with an additional cut (twice slot-split rectangular). The approach proposed here is not limited to these particular types of constructions and can be used for any other shapes.

Figure 1 shows the U-shaped microstrip resonator, in which the bandwidth is created by its two or three lowest oscillation modes [34,35]. The width and depth of the conductor cut influence the position of the lower natural frequencies, and tuning them allows bringing them closer. The bandwidth of a resonator can be determined as follows: (1)$$dF=\Delta F/F_{0}$$
where $\Delta F=F_{h}-F_{l}$, and $F_{h}$ and $F_{l}$ are the high-frequency and low-frequency boundaries of the bandwidth, measured at a −3 dB level from minimal losses, and $F_{0}$ is the center frequency of the bandwidth. Some of the previous studies on three-mode resonators of such construction have shown that the minimum bandwidth is around 45%, and the maximum is 71% when $F_{0}=3$ GHz [35]. Such structure is widely used due to its simplicity, and it finds application in various types of devices, such as sensors [3,36,37]. The results of the numerical electrodynamic modeling of such a resonator on a TBNS dielectric plate (thickness h=1 mm, relative permittivity ε=80) are very close to the parameters of the manufactured prototype [35].

Figure 1 also shows the geometric characteristics of the U-shaped 3-mode resonator. In particular, there are 13 parameters, set to the following values:$x_{1}=3.50$ mm—length of left side port connection;$x_{2}=0.90$ mm—width of the left rectangle;$x_{3}=2.20$ mm—width of the center rectangle;$x_{4}=0.90$ mm—width of the right rectangle;$x_{5}=3.50$ mm—length of right side port connection (always equal to $x_{1}$);$y_{1}=0.98$—relative height of the left side port connection;$y_{2}=0.02$ mm—width of the port connections;$y_{3}=2.80$ mm—distance from the bottom edge of the dielectric plate;$y_{4}=18.90$ mm—height of the left rectangle;$y_{5}=12.80$ mm—height of the center rectangle;$y_{6}=18.90$ mm—height of the right rectangle;$y_{7}=0.98$—relative height of the right side port connection;$y_{8}=2.80$ mm—distance to the top edge of the dielectric plate.

It is worth mentioning that unlike previous work [38], where the left- and right-hand sides of the resonator were symmetric and mirroring was applied, here, we allow asymmetric geometry. This leads to more parameters to be set but allows wider investigation of the proposed approach, as it can be further applied to design diplexers and multiplexers. From the 13 parameters listed above, only 8 are tunable, which are the rectangles parameters $x_{2}$, $x_{3}$, $x_{4}$, $y_{4}$, $y_{5}$, $y_{6}$ and port positions $y_{1}$ and $y_{7}$. The parameters described here result in a resonator with $F_{0}=1.498$ GHz and ΔF=0.618, as shown in Figure 2.

The second resonator considered in this study is shown in Figure 3, and its AFC is presented in Figure 4. This is a twice slot-split rectangular microstrip resonator, capable of 6-mode operation. In this case, the theoretical investigations are in agreement with the characteristics, measured on a manufactured prototype, meaning that the experiments could be performed automatically using only electrodynamic modeling tools. Previous experiments with this type of structure have shown that at $F_{0}=3$ GHz, it allows one to reach a bandwidth of around 80%.

All 17 parameters of the second structure, their values, and meanings are listed below:$x_{1}=3.50$ mm—length of left-side port connection;$x_{2}=0.60$ mm—width of the first rectangle;$x_{3}=4.20$ mm—width of the second rectangle;$x_{4}=0.30$ mm—width of the third rectangle;$x_{5}=4.20$ mm—width of the fourth rectangle;$x_{6}=0.60$ mm—width of the fifth rectangle;$x_{7}=3.50$ mm—length of right-side port connection (always equal to $x_{1}$);$y_{1}=0.98$—relative height of the left-side port connection;$y_{2}=0.02$ mm—width of the port connections;$y_{3}=2.40$ mm—distance from the bottom edge of the dielectric plate;$y_{4}=28.00$ mm—height of the first rectangle;$y_{5}=21.40$ mm—height of the second rectangle;$y_{6}=15.70$ mm—height of the third rectangle;$y_{7}=21.40$ mm—height of the fourth rectangle;$y_{8}=28.00$ mm—height of the fifth rectangle;$y_{9}=0.98$—relative height of the right side port connection;$y_{10}=3.00$ mm—distance to the top edge of the dielectric plate.

From the 17 parameters listed above, only 12 are tunable, which are the rectangular parameters $x_{2}$, $x_{3}$, $x_{4}$, $x_{5}$, $x_{6}$, $y_{4}$, $y_{5}$, $y_{6}$, $y_{7}$, $y_{8}$ and port positions $y_{1}$ and $y_{7}$. The parameters described here result in a resonator with $F_{0}=1.832$ GHz and ΔF=0.957, as shown in Figure 4.

The two types of resonator structures are used in this study for experiments with automatic topology tuning. The purpose of the search process will be to reach a certain frequency range and bandwidth. It was mentioned that for the second example, the twice slot-split rectangular microstrip sensor, it is possible to reach 6-mode operation, but the initial solution will have only 4 modes, as shown in Figure 4. So, one of the goals of the optimization method is to find the desired number of modes. The solution evaluation method is described in further sections.

### 2.2. Continuous Optimization Methods

The topology of the constructions described above is controlled by a number of numerical values, and tuning them allows for achieving different frequency-selective properties. However, the evaluation of a particular set of parameter values is expensive, i.e., time-consuming, so an efficient optimization technique should be applied in order to obtain the constructions with desired properties.

In general, an optimization problem can be formulated as follows: given a target function f(x), find an optimal vector $x_{opt}$ so that
(2)$$x_{opt}=\underset{x}{\mathrm{argmin}}f\left(x\right),$$
$x\in R^{D}$, where *D* is the search space dimension, and $x_{j}\in \left[x_{j,min},x_{j,max}\right]$ with $x_{j,min}$ and $x_{j,max}$ are the lower and upper boundaries.

The field of optimization methods nowadays is vast and includes a variety of approaches, depending on the problem. In the case considered in this study, the derivative information is not available, so a zero-order method that requires only target function values should be used. The existing derivative-free optimization techniques could be divided into local and global search methods; the former may be more cost-efficient but often becomes stuck in local optimum, whereas the latter have better exploration capabilities at the cost of additional target function evaluations. An efficient optimization tool should combine the advantages of both of these approaches.

### 2.3. Surrogate-Based Optimization

Solving a real-world engineering problem usually requires building a mathematical model of the considered system. If the theoretical knowledge about the system is available and usable, then the model can be built using it; however, in many cases, this is not applicable. When dealing with complex optimization problems, the only possible way of building a mathematical model could be through experimental data. The earliest studies have relied on the design of the experimental approach [39,40], but a more recent approach is to apply Bayesian methods, such as Gaussian processes (GPs), also known as kriging [41]. Based on the idea of applying GPs, the Efficient Global Optimization (EGO) algorithm was proposed [26]. In EGO, the built model is used as a surrogate, i.e., the optimum is found using it, and it becomes the new point to be evaluated. As kriging allows evaluating not only the mean μ but also the standard deviation σ in each point, in EGO, it is possible to use several types of criteria to determine the next point:Surrogate-Based Optimization (SBO), using only mean values μ;Lower Confidence Bound (LCB), using 3σ confidence interval, i.e., μ−3σ;Expected Improvement (EI), using the function $E\left[I\left(x\right)\right]=E\left[max\left(f_{min}-Y,0\right)\right]$, i.e., combining the information about the best point, μ and σ.

Here, *Y* is a random variable with normal distribution and parameters μ and σ, $f_{min}$ is the best-known function value, and *E* is the expectation. One of the disadvantages of EGO is that generating each new point requires solving another optimization problem using the surrogate model, and training the surrogate on the available data becomes more and more time-consuming as the dataset size and problem dimension grow. Nevertheless, the EGO approach has found many applications and proved to be an efficient search method for complex problems [42].

### 2.4. Differential Evolution

The evolutionary algorithms are optimization techniques, inspired by the process of natural selection. The earliest versions of EAs, such as genetic algorithms, have been shown to perform well on a variety of global optimization problems in different domains, including binary, combinatorial, integer, and numeric optimization [43]. There have been many other classes of methods proposed besides GAs, for example, evolutionary strategies (ES) and differential evolution (DE) [44]. The latter today is one of the most efficient and widely used global numeric optimizers due to simplicity and high efficiency [45]. In the latest optimization competitions of the Congress on Evolutionary Computation (CEC), the DE-based approaches occupy not just the leading positions but most of the submitted algorithms [46,47].

Differential evolution is a population-based optimizer, which means that it uses a set of *N* randomly generated vectors $x_{i,j}$, i=1,…,N, j=1,…,D, where *D* is the problem dimension. The main operation in DE is the difference-based mutation, one of the most widely used strategies is called current-to-pbest/1 [48]:(3)$$v_{i,j}=x_{i,j}+F\left(x_{pbest,j}-x_{i,j}\right)+F\left(x_{r1,j}-x_{r2,j}\right).$$
where $x_{i}$ is the *i*-th vector in the population of *N* individuals, and each vector consists of *D* variables; $v_{i}$ is called the donor vector; r1 and r2 are random indices from the population; pbest is one of the p% best individuals; and *F* is called the scaling factor. This mutation consists of two parts, exploitation, i.e., moving toward one of the p% best solutions with the first difference, and exploitation, that is, moving in a random direction with the second difference. This allows one to use both trends in DE.

Mutation is followed by crossover; the most popular method is the binomial crossover, implemented as follows:(4)$$u_{i,j}=\left\{\begin{array}{cc} v_{i,j}, & \mathrm{if}\;rand(0,1)\leq Cr\;\mathrm{or}\;j=jrand\\ x_{i,j}, & \mathrm{otherwise} \end{array}\right.,$$
where $u_{i}$ is the trial vector, jrand is required to make sure that at least one component is taken from the donor vector, and Cr is the crossover rate parameter. The generated trial vectors are also checked to be within the bounds and corrected if needed [49].

The trial solutions $u_{i}$ are then evaluated and compared with the target ones in the selection step:(5)$$x_{i,j}=\left\{\begin{array}{cc} u_{i,j}, & \mathrm{if}\;f\left(u_{i}\right)\leq f\left(x_{i}\right)\\ x_{i,j}, & \mathrm{if}\;f\left(u_{i}\right)>f\left(x_{i}\right) \end{array}\right..$$

That is, if the newly generated solution is better, it replaces the old one.

All modern DE variants rely on one of the parameter adaptation techniques. The reason for this is that the three main parameters, population size *N*, scaling factor *F*, and crossover rate Cr, significantly influence performance [50]. As the DE is designed to be a general-purpose black-box optimizer, the best way to set the parameters is to determine them based on how well certain values perform during the search. For the last 10 years, one of the most efficient parameter adaptation techniques for DE is the successful history-based adaptation (SHA), proposed in [51]. The L-SHADE algorithm [52], which tunes *F* and Cr with SHA and linearly decreases the population size, is a baseline approach for many modern DE variants [46].

Recently, there have been several attempts to change the general scheme of L-SHADE, for example, in the L-NTADE algorithm [47], where two populations are used. The successful history-based adaptation was shown to be biased [53], and a hyperheuristic approach was applied to design the parameter adaptation method automatically [54]. The algorithm proposed in this study is based on these results.

### 2.5. Previous Work on Microstrip Multimode Resonator Design

In our previous study [38], the same problem of microstrip multimode resonator topology design was considered. In that work, three types of constructions were considered: 3-mode, 5-mode, and 6-mode resonators. The main task was the same: given a baseline topology, tune its bandpass to the desired frequency range while keeping the same width. To optimize the topology, several approaches were considered, namely simulated annealing (SA), continuous genetic algorithm (CGA), and a hybrid between the genetic algorithm and differential evolution with parameter adaptation. A special fitness function was designed to evaluate the solutions, combining information about the number of modes, reflection losses, and the operating frequency range. Also, a heuristic approach was adopted to obtain an approximation of the desired solution given the initial and desired frequency—this was performed by scaling the whole topology.

The drawbacks of that study were that, in fact, the local search was performed around the scaled solution, and the mesh was fixed during the simulations. The computational resource was limited to 500 function evaluations. That is, although it was shown that it is possible to tune resonators to desired characteristics with this approach, it is not general. For example, if instead of tuning to a different frequency the aim is to increase the width of the passband, the scaling would not help, and this would require solving a global optimization problem. The methods used to overcome these limitations are described in the next section.

### 2.6. Proposed Approach: Target Function Formulation

The main focus of this story is on the development of an optimization technique, which would be general and capable of tuning different types of microwave structures. To achieve this, first of all, we determine the target function, also known as the fitness function in the evolutionary computation community. The target function will be used to determine the quality of the solution, i.e., its difference from the desired one, with a single numeric value. In the previous study, the quality of the solutions was determined with five different metrics, which were then combined into a single one [38]. Here, we follow a similar approach, but the metrics are changed to have the same scale—our preliminary experiments showed that this is crucial for obtaining high-quality results.

Let us suppose that the goal is to tune the resonator so that it would have the following two values determining the bandwidth:Low-frequency $F_{l}^{des}$;High-frequency $F_{u}^{des}$.
where “des” stands for “desired”. Hence, the width can be calculated as follows:(6)$$dF^{des}=\frac{F_{u}^{des}-F_{l}^{des}}{0.5(F_{u}^{des}+F_{l}^{des})}$$

The denominator is also called the center frequency, $F_{0}^{des}=0.5\left(F_{u}^{des}+F_{l}^{des}\right)$, and the nominator is $\Delta F^{des}=F_{u}^{des}-F_{l}^{des}$. The same set of values is calculated for the actual topology based on the results of electrodynamic modeling. These actual values are $F_{l}^{act}$, $F_{u}^{act}$, $F_{0}^{act}$, $\Delta F^{act}$, $dF^{act}$.

The optimization has the following five objectives:The desired and actual center frequencies of bandpass should be the same, $F_{0}^{des}=F_{0}^{act}$;The desired and actual bandwidth should be the same, $dF^{des}=dF^{act}$;The number of actual modes should be greater or equal to the desired ones, $NM^{act}\geq NM^{des}$;The frequency dependence of losses in reflection should not exceed the threshold level within the bandwidth, i.e., the actual losses threshold should be $LT^{act}<LT^{des}$;The width at the threshold level $LT^{des}$ should be greater than the scaled desired width, i.e., $\frac{F_{u,T}^{act}-F_{l,T}^{act}}{F_{0}^{des}}>CT\cdot dF^{des}$.

Here, $F_{u,T}^{act}$ and $F_{l,T}^{act}$ are the frequencies where the reflection losses hit the $LT^{des}$ level, and CT∈(0,1] determines the allowed narrowness at this level compared with $dF^{des}$. The $LT^{des}$ was set to −14 dB in all experiments. For the last objective, we determine
(7)$$F_{l,T}^{des}=F_{l}^{des}+\frac{CT}{2}\cdot \left(F_{u}^{des}-F_{l}^{des}\right)$$
(8)$$F_{u,T}^{des}=F_{l}^{des}-\frac{CT}{2}\cdot \left(F_{u}^{des}-F_{l}^{des}\right)$$

Note that the three last objectives could also be formulated as constraints. The mathematical formulation of all five objectives is given below.

$fit_{1}=\left|1-\frac{F_{0}^{act}}{F_{0}^{des}}\right|$;$fit_{2}=\left|1-\frac{dF^{act}}{dF^{des}}\right|$;$fit_{3}=\frac{NM^{des}-NM^{act}}{NM^{des}}$ if $NM^{act}<NM^{des}$, otherwise 0;$fit_{4}=\frac{LT^{act}}{LT^{des}}$ if $LT^{act}>LT^{des}$, otherwise 0;$fit_{5}=1-\frac{F_{u,T}^{act}-F_{l,T}^{act}}{F_{u,T}^{des}-F_{l,T}^{des}}$ if $\frac{F_{u,T}^{act}-F_{l,T}^{act}}{F_{u,T}^{des}-F_{l,T}^{des}}<1$, otherwise 0;

Also note that in the fourth objective, both $LT^{act}$ and $LT^{des}$ are negative, and, for example, $LT^{act}=-13$ is penalized. Finally, the fitness of a solution is just a sum of all five:(9)$$fit=fit_{1}+fit_{2}+fit_{3}+fit_{4}+fit_{5}$$

An example of calculating all of these values will be given in the experimental section.

### 2.7. Proposed Approach: Differential Evolution Modification

As mentioned above, in this study, a combination of the evolutionary algorithm, namely a variant of differential evolution, and surrogate-based optimization is proposed. First, the used DE algorithm is described, and then its hybrid with kriging is as well.

The DE algorithm proposed here is based on several works on parameter adaptation techniques for differential evolution. In particular, as DE is highly sensitive to setting the scaling factor *F*, in [53], it was shown that the existing successful history-based adaptation can be modified by biasing generated *F* to higher values. However, it is possible to design adaptation techniques automatically using genetic programming, as shown in [54]. Based on the findings of the paper in [55,56], a hyperheuristic approach to automatically design parameter adaptation techniques was implemented with a simple rule for tuning scaling factor *F* based on success rate.

The success rate SR is determined by the ratio of the number of improved solutions NS to the current population size *N*:(10)$$SR=\frac{NS}{N}$$

The number of improved solutions NS is calculated as the number of times when the selection (Equation (Equation 5)) step was successful. In particular, the mean for sampling scaling factor *F* values can be calculated as the *c*-th order root of the success rate [57]:(11)$$MF=SR^{1/c}$$
where c=4 is a typical setting of the parameter. Finally, the sampling for *F* is performed as follows:(12)F=randc(MF,0.1)
where randc(m,s) is a Cauchy-distributed random number with location parameter *m* and scale parameter *s*. If the sampled *F* value is smaller than 0, it is sampled again, and if it is larger than 1, it is set to 1.

The proposed algorithm called SRDE (Success-Rate-based Differential Evolution) also uses the external archive, the same as in the SHADE [51] algorithm. The first step is the initialization, which is performed based on the given baseline solution $b_{j}$ (parameters from Section 2.1 are used here). In particular, the population $x_{i,j}$ and the archive $a_{i,j}$ (i=1,2,…N, j=1,2,…D) are generated with normal distribution as follows:(13)$$x_{i,j}=randn\left(b_{j},0.05\cdot \left(xmax_{j}-xmin_{j}\right)\right)$$
where randn(m,s) is a normally distributed random variable with mean *m* and standard deviation *s*, and $xmax_{j}$, $xmin_{j}$ are the upper and lower bounds for variable *j*. In other words, the scaled range $0.05\cdot (xmax_{j}-xmin_{j})$ is used as the standard deviation. The archive $a_{i,j}$ is sampled in the same way.

The mutation strategy relied on selective pressure, which changes the probabilities of individuals being chosen for mutation based on their fitness, as described in [58]. In particular, the exponential selective pressure was applied to the r1 index in Equation (Equation 3). The rank of a particular individual in an array sorted by fitness is calculated as follows:(14)$$rank_{i}=\frac{-kp\cdot i}{N}$$
where kp is the selective pressure parameter. The resulting probabilities are calculated as $p_{i}=\frac{rank_{i}}{\sum_{j}^{N}rank_{j}}$.

After initialization, the main cycle begins. *F* values are generated as described above, and the crossover rate parameter is fixed as Cr=0.9. The mutation strategy used is the current-to-pbest with archive, the same as in Equation (Equation 3), but the last vector $x_{r2}$ can be chosen from the joined set of population and archive. After binomial crossover (Equation (Equation 4)), the solutions are checked to be within the search bounds using the midpoint-target method:(15)$$u_{i,j}=\left\{\begin{array}{cc} \frac{xmin_{j}+x_{i,j}}{2}, & \mathrm{if}\;v_{i,j}<x_{lb,j}\\ \frac{xmax_{j}+x_{i,j}}{2}, & \mathrm{if}\;v_{i,j}>x_{ub,j} \end{array}\right..$$

During the selection step, the solutions, which are replaced in the population, are put into the archive, replacing randomly chosen ones. Also, the number of successful replacements is calculated, as well as the success rate SR. This completes the single generation (iteration) of SRDE.

### 2.8. Proposed Approach: Surrogate-Assisted SRDE

One of the goals of the current study is to combine the fast convergence of the DE algorithm with the explorative properties of surrogate-based optimization. In particular, the hybrid approach, called surrogate-assisted SRDE (SA-SRDE), is proposed. In this algorithm, the basic algorithm is the SRDE, but in addition, the kriging model is built using the evaluated points, and every generation, one of the worst points in the SRDE population is replaced by the point found on the surrogate model.

The main steps of the proposed algorithm can be described as follows:Initialize SRDE algorithm (Equation (Equation 13)) and evaluate fitness (Equation (Equation 9)) $fit(x_{i})$, i=1,2,…N;Place current population $x_{i}$ and fitness values $y_{i}=fit\left(x_{i}\right)$ into dataset $\left[x_{i}^{ds},y_{i}^{ds}\right]$;Generate mutant vectors $v_{i}$ (Equation (Equation 3)) and trial vectors $u_{i}$ (Equation (Equation 4)), apply bound constraints (Equation (Equation 15));Evaluate fitness of trail vectors $y_{i}=fit\left(u_{i}\right)$, perform selection (Equation (Equation 5)), and add improved points $\left[u_{i},y_{i}\right]$ to the dataset, i=1,2,…M;Train a surrogate model $KRG(\left[x_{i}^{ds},y_{i}^{ds}\right])$ on the dataset;Apply acquisition function using best known solution $x_{b}$ to obtain $x_{af}=AF\left(KRG,x_{b}\right)$ and evaluate fitness $y_{af}=fit\left(x_{af}\right)$;Replace the worst individual $x_{w}$ in the population with $x_{af}$ if $y_{af}<y_{w}$;If the termination condition is not met, go to step 2;Return best solution $x_{b}$, $y_{b}$.

The size of the dataset *M* may differ depending on the number of improved points. Step 6 requires additional explanation, as it contains the acquisition function, which is another SRDE algorithm applied to the surrogate model. The search range is set around the best-known solution $x_{b}^{input}$. The steps of $AF(KRG,x_{b}^{input})$ are described below:Use the best-known solution to set the temporary search boundaries: $xmin_{j}=x_{b}^{input}-0.05\cdot \left(xmax_{j}-xmin_{j}\right)$, $xmax_{j}=x_{b}^{input}+0.05\cdot \left(xmax_{j}-xmin_{j}\right)$, j=1,2,…D;Initialize the SRDE algorithm (Equation (Equation 13)) and evaluate fitness using the LCB approach (Section 2.3): $fit\left(x_{i}\right)=\mu \left(x_{i}\right)-3\sigma \left(x_{i}\right)$, i=1,2,…N;Generate mutant vectors $v_{i}$ (Equation 3)) and trial vectors $u_{i}$ (Equation (Equation 4)), apply bound constraints (Equation (Equation 15));Evaluate fitness of trail vectors $y_{i}=fit\left(u_{i}\right)=\mu \left(u_{i}\right)-3\sigma \left(u_{i}\right)$ and perform selection (Equation (Equation 5));If the termination condition is not met, go to step 2;Return best solution $x_{b}$, $fit(x_{b})$.

In Figure 5, the flowchart of the proposed approach is shown. If the surrogate is not used (SRDE), only the left part of the scheme is active; otherwise, both are active (SA-SRDE).

Instead of the LCB approach, other variants, such as SBO or EI, could be applied. Note that the EGO algorithm uses the SLSQP approach to perform optimization on the surrogate model, but a combination of LCB and SRDE has shown better results in preliminary experiments.

In the next section, the details of the performed computational experiments are given, including parameter settings of the used algorithms, as well as the final results, approach comparison, and the applicability of the results for constructing various devices.

## 3. Results and Discussion

The number of experiments in this study is mainly limited by the computational complexity of the electrodynamic modeling of the resonators. Evaluating a single set of parameters may take several minutes, so finding a topology with acceptable characteristics may take several days. The optimization algorithm was developed in Python, and the Surrogate Modeling Toolbox (SMT) [59,60] was applied for EGO and kriging. The experiments ran on six computers with Intel Core i9 13900 processors and 32Gb RAM.

### 3.1. Experimental Set-Up

There were two topologies considered (see Section 2.1), and several experiments were performed for each of them. In particular, the following tests were conducted:EGO algorithm on the first topology with recommended parameters from SMT;Standard SRDE algorithm on the first topology;SA-SRDE algorithm with LCB-based acquisition function on the first topology;Standard SRDE algorithm on the second topology;SA-SRDE algorithm with LCB-based acquisition function on the second topology;

For the first structure, the following required parameters were set: $F_{l}^{des}=1.5$ GHz and $F_{u}^{des}=3.0$ GHz, and for the second structure, $F_{l}^{des}=1.2$ GHz and $F_{u}^{des}=3.2$ GHz. During every experiment, all the evaluated parameters were stored for later analysis, as well as the time intervals to evaluate solutions and run the algorithm. The search ranges for every variable for the first and second topologies, as well as the baseline solution xbas parameters, are shown in Table 1 and Table 2.

### 3.2. Results on the First Structure

Table 1 and Table 2 only show the parameters, which were actually tuned during the optimization process. The ranges were set with enough margin so that the search algorithm would not get stuck because of them.

The EGO algorithm was used with the following settings: initial number of points: 25; total number of target function evaluations (simulations): $NFE_{max}=1000$; and criterion: expected improvement (EI). The parameters of the SRDE and SA-SRDE shared the same parameters regarding the differential evolution part, and they are listed below:Computational resource $NFE_{max}=1000$ evaluations for first structure, $NFE_{max}=1500$ for second structure;Number of independent runs NR=15 for first structure, NR=10 for second structure;Population size N=25 for first structure, N=36 for second structure;Mutation parameter p=0.3 (Equation (Equation 3));Crossover rate parameter Cr=0.9 (Equation (Equation 4));Scaling factor calculation parameter c=4 (Equation (Equation 11));Selective pressure parameter kp=3 (Equation (Equation 14)).

The best target function values for each of the 15 runs in the first three experiments are shown in Table 3.

As can be seen from Table 3, the EGO algorithm was not able to achieve good results in most of the runs; however, both SRDE and SA-SRDE performed much better. SA-SRDE achieved the best mean, standard deviation, median, and maximum values but obtained the same result as SRDE in terms of the best found solution out of 15 runs. The best values in Table 3 are shown in bold. Table 4 contains the statistical analysis of the results, shown above.

The Student’s *t*-test and Mann–Whitney *U* test show that both SRDE and SA-SRDE outperform EGO; however, the comparison between SRDE and SA-SRDE does not show any statistically significant differences between these two. Nevertheless, as Table 3 has shown, SA-SRDE is better on average.

Figure 6 shows the convergence graphs of the three tested algorithms. The wide lines are the average over all runs.

As Figure 6 demonstrates, the EGO algorithm performs much worse than the SRDE and SA-SRDE. Comparing SRDE and SA-SRDE, it can be noted that after 300 evaluations, the SA-SRDE starts to show better results on average.

### 3.3. Results on the Second Structure

The results of the second set of experiments with the second structure are shown in Table 5 and Table 6 contains the statistical comparison. The experiments with EGO on the second structure were not performed, because this method showed poor results in the first set of tests. The best values in Table 5 and Table 6 are shown in bold.

As Table 5 shows, the SA-SRDE outperformed the baseline approach in most of the basic statistical characteristics. The statistical tests also show very small *p*-values, around $8.7\times 10^{-2}$, meaning that the difference is rather significant.

Figure 7 is similar to Figure 6 and shows the convergence graphs of the two tested algorithms for the second structure. The wide lines are the average over all runs.

Figure 7 demonstrates that SA-SRDE outperforms SRDE after 500 evaluations and achieves better results on average at the end.

### 3.4. Time Requirements Analysis

Considering the analysis above, we may conclude that the SA-SRDE performs better than the baseline approach SRDE due to the usage of the surrogate approach. However, it is important to consider that the SA-SRDE requires additional computational effort in order to build models and optimize them. The time requirements in seconds of these two methods are shown in Table 7 and Table 8.

The comparison of time, required to finish the optimization in Table 7 and Table 8, demonstrates that the SA-SRDE does not always work longer. In particular, for the first structure, the SA-SRDE worked around 10% longer than the baseline algorithm SRDE on average, but for a more complex second structure, the times of SA-SRDE are smaller. This can be explained by the fact that the tuning is performed towards higher frequency ranges, and it results in smaller structures in terms of geometric characteristics. Smaller structures tend to require less computational resources to be modeled, so as SA-SRDE performs better and replaces worse solutions, it generally evaluates smaller structures more often, resulting in an advantage in computation time, as Table 8 shows. However, this advantage would disappear if the goal would be to tune to lower frequency ranges.

### 3.5. Analysis of Best Found Solutions

To show the quality of the best found solutions in this section the best solutions found for both structures are shown. Figure 6 shows the first structure, optimized with SA-SRDE, 15-th run, and Figure 6 shows its AFC. Table 9 contains the geometric parameters.

Figure 5 shows that the optimization process results in an asymmetric structure of the resonator. The proposed approach allows one to obtain full symmetry, but it requires increased computational resources. The AFC characteristics of the found solution are almost ideal: the frequencies are very close to the desired ones, and the $S_{11}$ curve does not reach the −14 dB level.

Figure 8 and Figure 9 show the geometry and AFC of the optimized second structure from run 2 of SA-SRDE, and Table 10 contains the geometric parameters.

Unlike the first structure, here, in Figure 10, it can be seen that the topology is almost perfectly symmetric, and the values in Table 10 support this statement. As for the AFC in Figure 11, unlike the one shown in Figure 4, here, we may clearly observe six modes, and the frequency range is even larger than the required 1.2 and 3.2 GHz. It shows that there is a potential room for further improvement of this variant.

## 4. Conclusions

In this paper, a complex engineering optimization problem of tuning the conductor’s topologies of the microstrip multimode resonators was considered. This task is characterized by high computational complexity, as it requires electrodynamic modeling of the microwave devices, which takes minutes of time even on modern hardware. To solve this problem a hybrid surrogate-assisted differential evolution algorithm was proposed and tested on the problem of tuning two resonators with different topologies. The experiments have shown that this approach outperforms the baseline algorithm without the kriging model and allows one to find the efficient topologies of microwave devices.

It is important to mention that the approach proposed in this study is not limited to the design of microstrip resonators and can be utilized in other areas of complex engineering design, where the numeric parameters should be tuned. In the area of microwave devices, the algorithm can be applied to design diplexers, multiplexers, or power dividers, and the only things that should be changed are the search ranges and fitness calculation. As for developing more efficient search algorithms, unfortunately, it is difficult to search for more efficient optimizers for this type of expensive problem, so further developments should be performed on simpler problems that have similar landscape properties to the problems considered in this study. Determining these properties could be one of the directions of further studies.

## Figures and Tables

**Figure 1 sensors-24-05057-f001:**
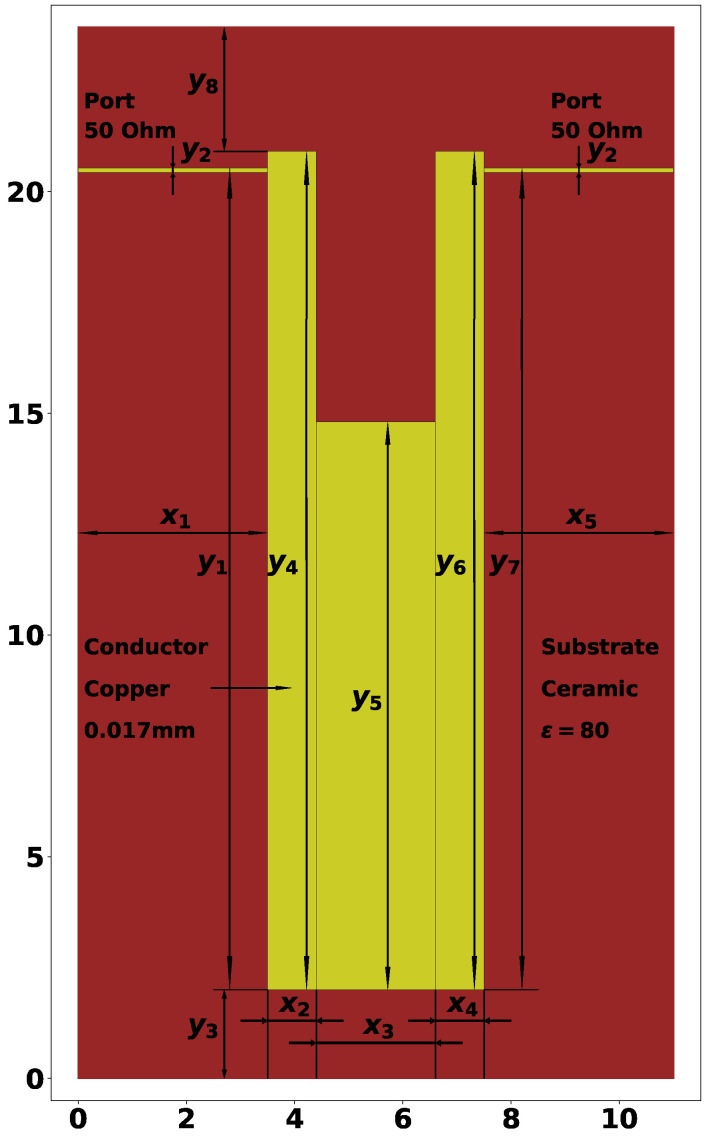
U-shaped (slot-split rectangular) microstrip resonator and its main geometric parameters.

**Figure 2 sensors-24-05057-f002:**
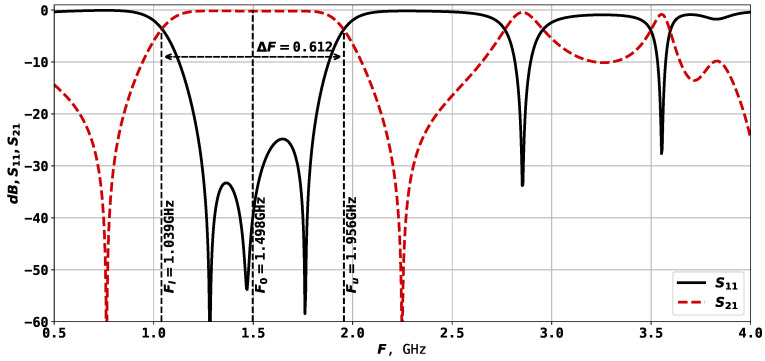
AFC of the U-shaped microstrip resonator.

**Figure 3 sensors-24-05057-f003:**
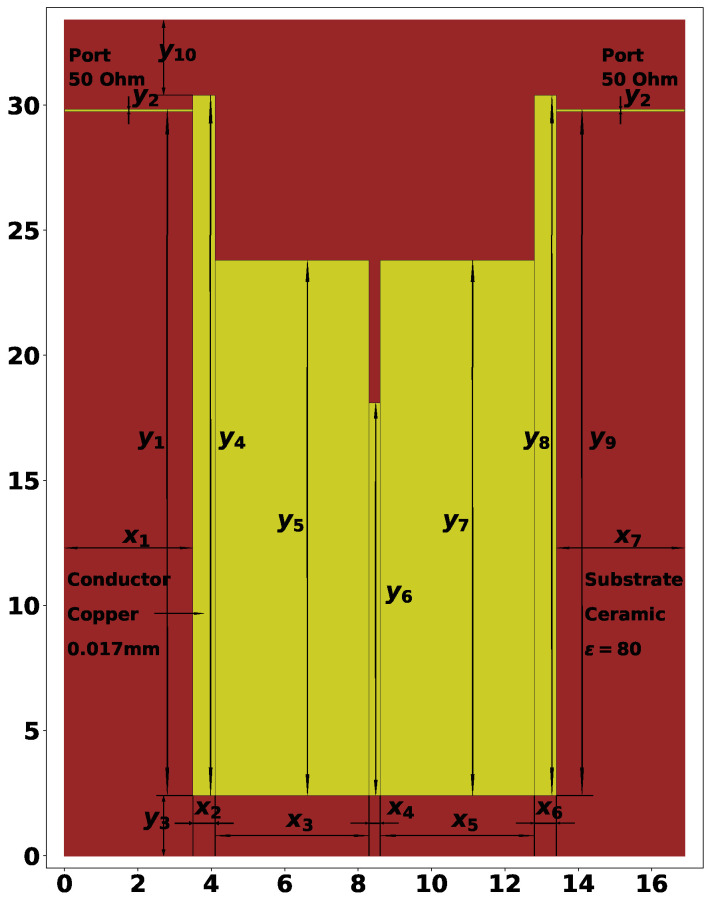
U-shaped (twice slot-split rectangular) microstrip resonator and its main geometric parameters.

**Figure 4 sensors-24-05057-f004:**
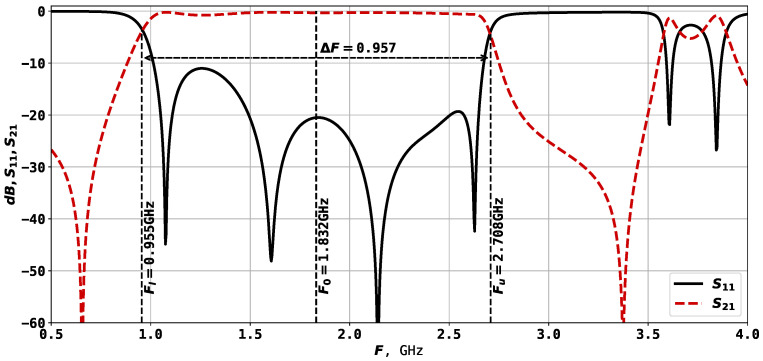
AFC of the U-shaped microstrip resonator with an additional cut.

**Figure 5 sensors-24-05057-f005:**
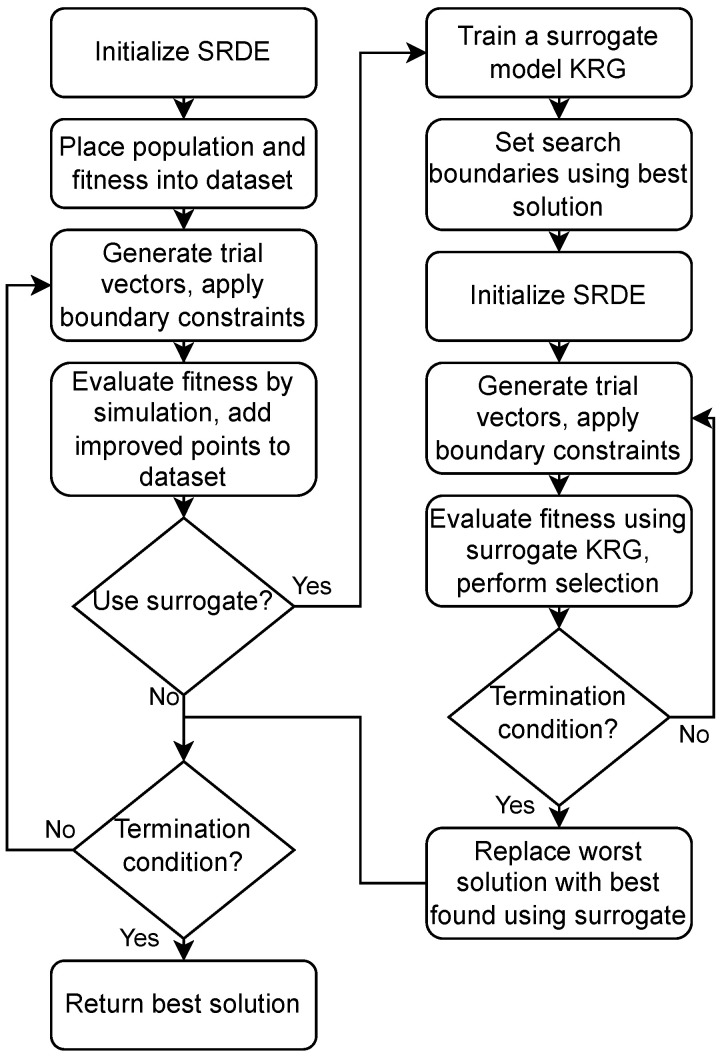
Flowchart of the proposed SRDE/SA-SRDE algorithm.

**Figure 6 sensors-24-05057-f006:**
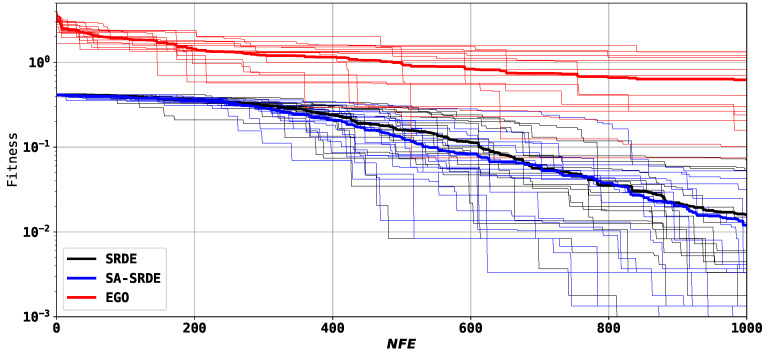
Convergence graphs of EGO, SRDE, and SA-SRDE; first structure.

**Figure 7 sensors-24-05057-f007:**
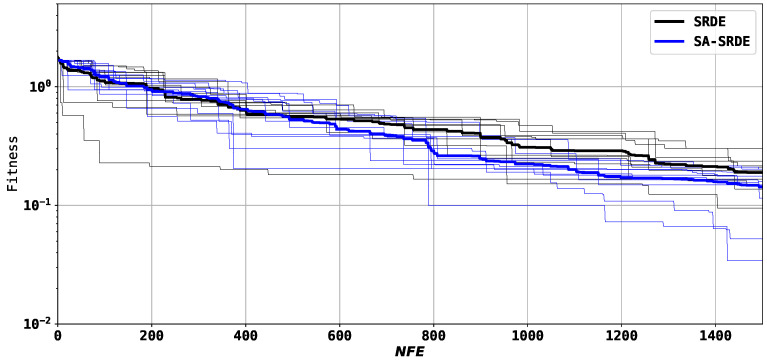
Convergence graphs of SRDE and SA-SRDE; second structure.

**Figure 8 sensors-24-05057-f008:**
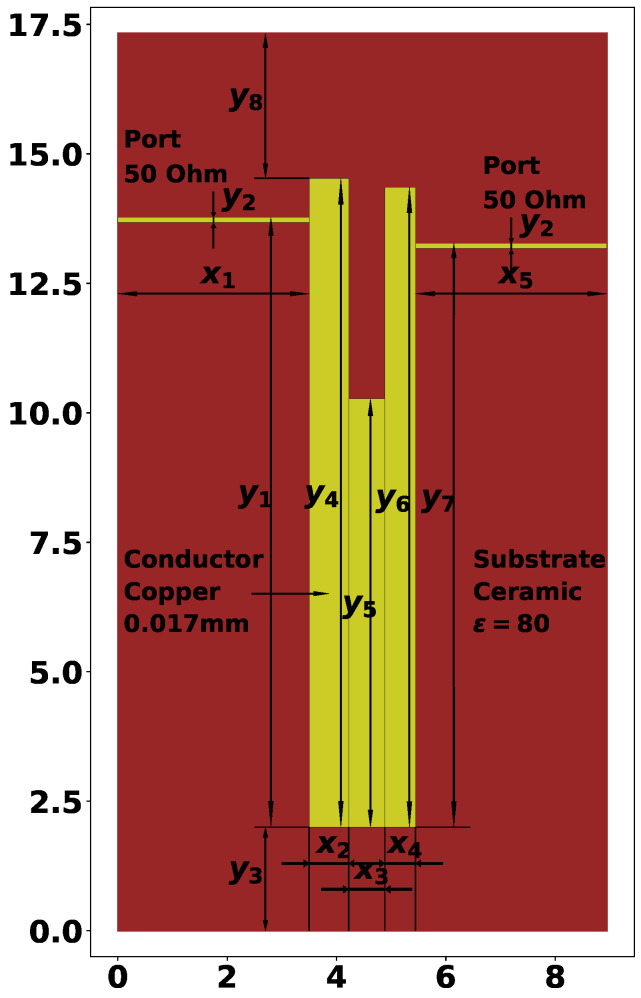
Optimized U-shaped microstrip resonator.

**Figure 9 sensors-24-05057-f009:**
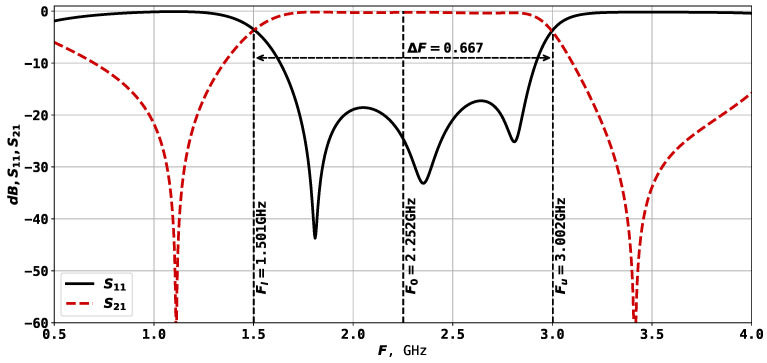
AFC of the optimized U-shaped microstrip resonator.

**Figure 10 sensors-24-05057-f010:**
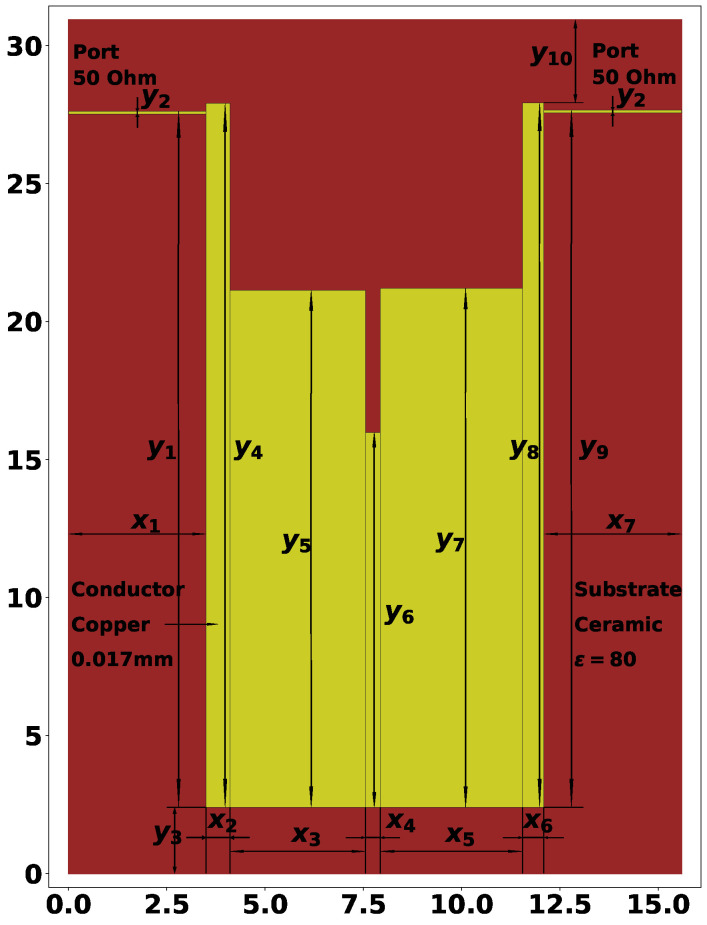
Optimized U-shaped microstrip resonator with additional cut.

**Figure 11 sensors-24-05057-f011:**
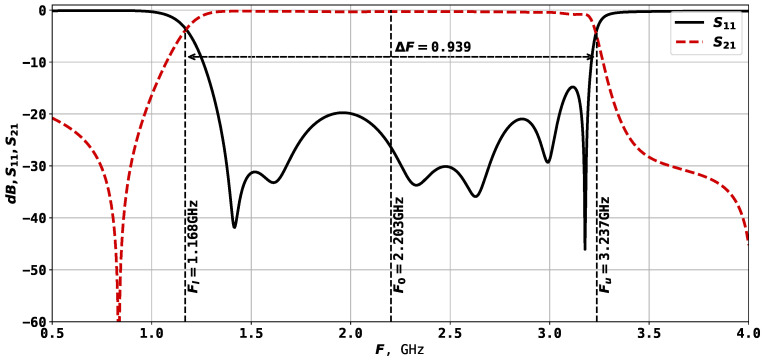
AFC of the optimized U-shaped microstrip resonator with additional cut.

**Table 1 sensors-24-05057-t001:** Search range and baseline solution for the first structure, in mm.

Parameter	${\mathrm{xmin}}_{j}$	${\mathrm{xbas}}_{j}$	${\mathrm{xmax}}_{j}$
$x_{2}$	0.30	0.90	3.00
$x_{3}$	0.30	2.20	5.00
$x_{4}$	0.30	0.90	3.00
$y_{1}$	0.00	0.98	1.00
$y_{4}$	1.00	18.90	30.00
$y_{5}$	1.00	12.80	30.00
$y_{6}$	1.00	18.90	30.00
$y_{7}$	0.00	0.98	1.00

**Table 2 sensors-24-05057-t002:** Search range and baseline solution for the second structure, in mm.

Parameter	${\mathrm{xmin}}_{j}$	${\mathrm{xbas}}_{j}$	${\mathrm{xmax}}_{j}$
$x_{2}$	0.01	0.60	2.00
$x_{3}$	0.01	4.20	10.00
$x_{4}$	0.01	0.30	2.00
$x_{5}$	0.01	4.20	10.00
$x_{6}$	0.01	0.60	2.00
$y_{1}$	0.00	0.98	1.00
$y_{4}$	0.10	28.00	40.00
$y_{5}$	0.10	21.40	40.00
$y_{6}$	0.10	15.70	40.00
$y_{7}$	0.10	21.40	40.00
$y_{8}$	0.10	28.00	40.00
$y_{9}$	0.00	0.98	1.00

**Table 3 sensors-24-05057-t003:** Results of experiments with the first structure.

Run	EGO	SRDE	SA-SRDE
1	$1.323\times 10^{0}$	$1.301\times 10^{-2}$	$1.333\times 10^{-3}$
2	$2.653\times 10^{-1}$	$7.050\times 10^{-2}$	$5.267\times 10^{-2}$
3	$1.044\times 10^{0}$	$1.333\times 10^{-3}$	$3.670\times 10^{-3}$
4	$2.067\times 10^{-1}$	$4.440\times 10^{-3}$	$9.998\times 10^{-4}$
5	$8.171\times 10^{-1}$	$5.308\times 10^{-2}$	$3.664\times 10^{-3}$
6	$1.206\times 10^{0}$	$3.333\times 10^{-3}$	$4.112\times 10^{-3}$
7	$3.020\times 10^{-1}$	$5.512\times 10^{-2}$	$1.298\times 10^{-2}$
8	$6.949\times 10^{-1}$	$3.331\times 10^{-3}$	$3.174\times 10^{-2}$
9	$7.497\times 10^{-2}$	$5.674\times 10^{-3}$	$1.662\times 10^{-2}$
10	$2.408\times 10^{-1}$	$9.998\times 10^{-4}$	$3.333\times 10^{-3}$
11	$4.060\times 10^{-1}$	$1.333\times 10^{-3}$	$4.252\times 10^{-2}$
12	$1.019\times 10^{-1}$	$4.440\times 10^{-3}$	$3.333\times 10^{-3}$
13	$1.339\times 10^{0}$	$6.007\times 10^{-3}$	$1.333\times 10^{-3}$
14	$1.592\times 10^{-1}$	$1.499\times 10^{-2}$	$9.998\times 10^{-4}$
15	$1.145\times 10^{0}$	$1.778\times 10^{-3}$	$9.998\times 10^{-4}$
Min	$7.497\times 10^{-2}$	$9.998\times 10^{-4}$	$9.998\times 10^{-4}$
Mean	$6.217\times 10^{-1}$	$1.596\times 10^{-2}$	$1.202\times 10^{-2}$
Std	$4.632\times 10^{-1}$	$2.242\times 10^{-2}$	$1.621\times 10^{-2}$
Median	$4.060\times 10^{-1}$	$4.440\times 10^{-3}$	$3.664\times 10^{-3}$
Max	$1.339\times 10^{0}$	$7.050\times 10^{-2}$	$5.267\times 10^{-2}$

**Table 4 sensors-24-05057-t004:** Statistical comparison of first three experiments.

Run	Student’s Test		Mann–Whitney Test	
	t	p	U	p
EGO vs. SRDE	4.887	$3.781\times 10^{-5}$	0	$1.687\times 10^{-6}$
EGO vs. SA-SRDE	4.921	$3.440\times 10^{-5}$	0	$1.675\times 10^{-6}$
SRDE vs. SA-SRDE	0.533	$5.984\times 10^{-1}$	93	$2.148\times 10^{-1}$

**Table 5 sensors-24-05057-t005:** Results of experiment with the second structure.

Run	SRDE	SA-SRDE
1	$9.472\times 10^{-2}$	$2.024\times 10^{-1}$
2	$1.865\times 10^{-1}$	$3.421\times 10^{-2}$
3	$1.664\times 10^{-1}$	$2.020\times 10^{-1}$
4	$3.021\times 10^{-1}$	$1.585\times 10^{-1}$
5	$2.361\times 10^{-1}$	$1.153\times 10^{-1}$
6	$1.373\times 10^{-1}$	$1.766\times 10^{-1}$
7	$2.152\times 10^{-1}$	$1.486\times 10^{-1}$
8	$1.480\times 10^{-1}$	$5.233\times 10^{-2}$
9	$2.103\times 10^{-1}$	$1.875\times 10^{-1}$
10	$1.972\times 10^{-1}$	$1.573\times 10^{-1}$
Min	$9.472\times 10^{-2}$	$3.421\times 10^{-2}$
Mean	$1.894\times 10^{-1}$	$1.435\times 10^{-1}$
Std	$5.476\times 10^{-2}$	$5.605\times 10^{-2}$
Median	$1.918\times 10^{-1}$	$1.579\times 10^{-1}$
Max	$3.021\times 10^{-1}$	$2.024\times 10^{-1}$

**Table 6 sensors-24-05057-t006:** Statistical comparison of the two last experiments.

Run	Student’s Test		Mann–Whitney Test	
	t	p	U	p
SRDE vs. SA-SRDE	1.757	$9.586\times 10^{-2}$	31	$8.099\times 10^{-2}$

**Table 7 sensors-24-05057-t007:** Time requirements of first three experiments in seconds.

Statistic	EGO	SRDE	SA-SRDE
Min	$1.931\times 10^{+5}$	$7.193\times 10^{+4}$	$8.234\times 10^{+4}$
Mean	$4.042\times 10^{+5}$	$9.148\times 10^{+4}$	$1.033\times 10^{+5}$
Std	$1.192\times 10^{+5}$	$1.754\times 10^{+4}$	$1.358\times 10^{+4}$
Median	$4.090\times 10^{+5}$	$8.763\times 10^{+4}$	$1.062\times 10^{+5}$
Max	$5.756\times 10^{+5}$	$1.509\times 10^{+5}$	$1.246\times 10^{+5}$

**Table 8 sensors-24-05057-t008:** Time requirements of the last two experiments in seconds.

Statistic	SRDE	SA-SRDE
Min	$2.117\times 10^{+5}$	$2.051\times 10^{+5}$
Mean	$2.855\times 10^{+5}$	$2.651\times 10^{+5}$
Std	$5.184\times 10^{+4}$	$3.011\times 10^{+4}$
Median	$2.824\times 10^{+5}$	$2.673\times 10^{+5}$
Max	$3.516\times 10^{+5}$	$3.083\times 10^{+5}$

**Table 9 sensors-24-05057-t009:** Optimized parameters of the first structure in mm.

Parameter	$x_{2}$	$x_{3}$	$x_{4}$	$y_{1}$	$y_{4}$	$y_{5}$	$y_{6}$	$y_{7}$
Value	0.725	0.660	0.561	0.940	12.528	8.276	12.357	0.912

**Table 10 sensors-24-05057-t010:** Optimized parameters of the second structure in mm.

Parameters	$x_{2}$	$x_{3}$	$x_{4}$	$x_{5}$	$x_{6}$	$y_{1}$
Values	0.603	3.446	0.377	3.621	0.543	0.989
Parameters	$y_{4}$	$y_{5}$	$y_{6}$	$y_{7}$	$y_{8}$	$y_{9}$
Values	25.505	18.739	13.584	18.799	25.527	0.990

## Data Availability

Data are contained within the article.

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
