# Peer review of "Surrogate-Assisted Differential Evolution for the Design of Multimode Resonator Topology"

_sensors, 2024, doi:10.3390/s24155057_

Round 1
Reviewer 1 Report
Comments and Suggestions for Authors
The authors proposed surrogate-assisted differential evolution optimization approach for designing bandpass filters using multimode resonator. The algorithm was proved to have high-quality solution at every iteration. However, there are some points needed to be reviewed.
1. The initial values of the parameters are so close to the optimal values. It especially depends on the experience. How to determine the boundaries and initial values of the parameters without experience?
2. For determining the optimization goal, how to calculate the number of actual modes of the resonator NMact?
3. The paper has many abbreviations, each one should have explanation, what is the meaning of NFEmax?
4. When the optimal structure of a resonator is unknow, the resonant frequencies and coupling between the modes and the feeding-lines are also unknow. How to know the bandwidth in advance and calculate the fitness function?
Author Response
Answers to Reviewer 1:
The authors proposed surrogate-assisted differential evolution optimization approach for designing bandpass filters using multimode resonator. The algorithm was proved to have high-quality solution at every iteration. However, there are some points needed to be reviewed.
Answer: Thank you for your valuable comments. We tried to do our best of answer them and correct the paper accordingly.
- The initial values of the parameters are so close to the optimal values. It especially depends on the experience. How to determine the boundaries and initial values of the parameters without experience?
Answer: In this study we consider the case when the general structure of the device is known, but the particular parameter values are not, and they should be tuned to meet the requirements. That is, given a tuned resonator for one frequency range, it is required to tune it to a different range. While it is true that the boundaries are set from some experience with resonator design in general, the search ranges are rather big, for example, from 1 to 30 for y4-y6 for first structure – this is done so that the bounds do not simplify the search for solution.
- For determining the optimization goal, how to calculate the number of actual modes of the resonator NMact?
Answer: The microstrip resonator has an infinite number of oscillation modes. However, in order to get the lowest of them closer together in terms of frequency and create a bandwidth using them the parameter tuning is required. So the increase of the number of used oscillation modes requires introducing new parameter values (bending conductors, splitting, them, adding connections and so on). For single-mode resonators there is an equation that connects the number of parameters with the filter order. But for multimode resonators it does not work, as it is a more complicated task.
- The paper has many abbreviations, each one should have explanation, what is the meaning of NFEmax?
Answer: We are sorry for the confusion. The NFEmax means the number of target function evaluations, available to the algorithm, in other words, the number of simulations. We’ve added this to the abbreviations list at the end of the paper.
- When the optimal structure of a resonator is unknow, the resonant frequencies and coupling between the modes and the feeding-lines are also unknow. How to know the bandwidth in advance and calculate the fitness function?
Answer: As we have mentioned in the answer to the first question, the goal is to tune the structure to the desired frequency range. This frequency range is given by the client/customer, interested in the design with particular characteristics. So, setting the lower and upper frequencies, e.g. 1.5GHz and 3.0 GHz, as well as losses level, e.g. -14dB, gives the required values to calculate fitness. The precise determination of the maximum and minimum bandwidth of multimode resonator filters can be performed only after AFC calculation. Also there are parameters, which influence it significantly: dielectric constant of the substrate, its thickness, the central frequency of the filter bandwidth and the number of used resonator modes. It is generally known that the relative width of single-mode resonators does not exceed 40%. The relative width of multimode resonators is significantly wider. The baseline width value can be set based on the experience with designing such structures.
Reviewer 2 Report
Comments and Suggestions for Authors
This paper proposes a method for automatic search for multi-mode resonators combined with evolutionary calculation methods and modeling. In this paper, a variant of the differential evolution optimizer is used to construct an objective function model. By using two microwave filter experiments, the feasibility of the proposed algorithm to solve the design problem of microwave filter with actual tuning structure is illustrated. The optimization method in this paper has theoretical and engineering application value in the design of microwave devices.
I suggest that the author should explain and revise the following issues in the paper:
1. In order to describe the optimization process in more detail, the authors may show the convergence curve of fitness in the paper.
2. A flowchart of the design methodology is needed, so that the principles can be presented more clearly.
3. The structure of the microstrip resonator should be depicted more clearly. The feeding ports, metal and substrate should be labelled.
4. It is recommended to add the advantages of comparing the design effect with the existing professional software such as HFSS,CST and ADS.
5. If the in-band ripple coefficient of the bandpass filters is required in the design, how does this method deal with it?
Author Response
Answers to reviewer 2:
This paper proposes a method for automatic search for multi-mode resonators combined with evolutionary calculation methods and modeling. In this paper, a variant of the differential evolution optimizer is used to construct an objective function model. By using two microwave filter experiments, the feasibility of the proposed algorithm to solve the design problem of microwave filter with actual tuning structure is illustrated. The optimization method in this paper has theoretical and engineering application value in the design of microwave devices.
Answer: Thank you for your valuable comments. We tried to do our best of answer them and correct the paper accordingly.
I suggest that the author should explain and revise the following issues in the paper:
- In order to describe the optimization process in more detail, the authors may show the convergence curve of fitness in the paper.
Answer: We have added the convergence graphs with averaged results in Figures 6 and 7.
- A flowchart of the design methodology is needed, so that the principles can be presented more clearly.
Answer: We have added the flowchart in Figure 5 in the updated manuscript.
- The structure of the microstrip resonator should be depicted more clearly. The feeding ports, metal and substrate should be labelled.
Answer: We have added the labeling to the figures, containing resonators. The conductor material was copper, with thickness of 0.017mm, the filter ports had resistance of 50 Ohm, the substrate material was ceramic with dielectric constant 80.
- It is recommended to add the advantages of comparing the design effect with the existing professional software such as HFSS,CST and ADS.
Answer: Unfortunately, we do not have access to any of the mentioned software since 2022, and will not have it in the observable future. In fact, this was one of the motivations of developing our own software and optimization tools. However, we have previously performed some experiments with the first structure in CST, and were not satisfied with the results. Applying the built-in filter designer tool with various optimizers, such as trust region framework or CMA-ES mainly resulted in high losses, up to -6dB even after 500 modeling steps (fitness evaluations). As those experiments were performed in different conditions, we believe that such comparison would be unfair, and thus we did not include any information about it in the manuscript. Some of our colleagues from other institutes currently have access to CST license, so we may ask them to perform the experiments for us, but it would take at least 2 weeks to get the results. Please let us know if you believe that this should be done.
- If the in-band ripple coefficient of the bandpass filters is required in the design, how does this method deal with it?
Answer: We are not sure if we understood the question correctly. Are you referring to the unevenness of electromagnetic waves passage within the bandwidth frequency? If so, it is controlled by the level of losses, which was set to -14dB in our experiments. If it should be decreased, we may set the level of losses to -20dB, -30dB, -50dB and so on. The proposed approach allows doing this. If this is not what you meant, we would ask you to clarify the question.